# Vitamin D: Can Gender Medicine Have a Role?

**DOI:** 10.3390/biomedicines11061762

**Published:** 2023-06-19

**Authors:** Tiziana Ciarambino, Pietro Crispino, Giovanni Minervini, Mauro Giordano

**Affiliations:** 1Internal Medicine Department, Hospital of Marcianise, ASL Caserta, 81037 Caserta, Italy; 2Internal Medicine Department, Hospital of Latina, ASL Latina, 04100 Latina, Italy; 3Emergency Department, Hospital of Lagonegro, AOR San Carlo, 85042 Lagonegro, Italy; 4Advanced Medical and Surgical Sciences Department, University of Campania, L. Vanvitelli, 81100 Naples, Italy; mauro.giordano@gmail.com

**Keywords:** gender differences, vitamin D, biological functions, estrogens, glucocorticoids, inflammatory cytokines

## Abstract

This narrative review aims to shed light on the role of gender differences, on the biological and molecular functions in the main pathological mechanisms that recognize the role of vitamin D. Vitamin D deficiency is widespread worldwide, but it is still very controversial whether the amount of vitamin D taken daily is actually the only problem related to its biological functions. Currently, the plasma concentration of 25-hydroxyvitamin D represents the only indicator of the circulating blood quota. The concept is that the biological function of vitamin D is not only linked to its circulating levels, but it is hypothesized that its biological functions depend, above all, on its total bioavailability. In particular, vitamin D circulates for the most part linked to albumin and vitamin D binding protein (DBP), which depend on various pathological conditions and physiologically, above all, the function of the latter is regulated by estrogens, glucocorticoids, and inflammatory cytokines. During her life, women undergo various changes in the hormonal and sexual sphere concerning menarche, possible pregnancies, and breastfeeding but also the use of contraceptives and, finally, the transition from the period of fertility to menopause. Each of these phases presents specific needs and, consequently, sometimes also specific criticalities. Studies on young women have shown that vitamin D deficiency is present in 58 to 91% of cases. Obesity, metabolic disorders, and variation in estrogen contraction may affect vitamin D deficiency due to the decreased bioavailability from dietary sources due to deposition in body fat compartments.

## 1. Background

It is customary to think that the biological role of vitamin D is particularly that of preventing and combating osteoporosis and sarcopenia, which are prevalent conditions in the general population and which, through a gradual process, tend to involve older people with complications ranging from the increased predisposition to bone fractures up to sarcopenia and immobilization syndrome [1]. However, it is useful to point out that vitamin D has a potential role of regulating many cellular functions. Considering its expression in many cell types and its common deficiency, it is linked to several health problems that go far beyond those related to the musculoskeletal system, including infections, autoimmune disorders, cardiovascular disease, metabolic syndrome, and related conditions to insulin resistance, and, last but not least, its deficiency is part of the genesis of some malignant tumors [1,2,3,4]. Furthermore, it has been reported that vitamin D can be a powerful mediator of inflammation and, therefore, it would be part of the genesis of some respiratory pathologies of the gastrointestinal and genitourinary tracts [5]. It is obvious to think that due to the importance of the role that vitamin D plays in our organism, it is necessary to have levels of vitamin D. It is equally necessary, however, to understand if, in addition to the daily intake of vitamin D, there are alterations that concern one’s own metabolism and that can modulate its bioavailability. The synthesis of vitamin D3 or cholecalciferol begins following exposure of the skin to ultraviolet radiation. 7-dehydrocholesterol or provitamin D is present in the basal layer and in the spinous layer of the epidermis and undergoes conversion into pre-vitamin D with exposure to ultraviolet B rays and is then subsequently isomerized into vitamin D. Vitamin D2, also called ergocalciferol, derives from food sources of a vegetable nature and is, therefore, absorbed in the intestine. Thus, most of the vitamin D originates from the heal and passes through the capillaries of the dermis carried mainly by vitamin D binding protein (DBP) and albumin to the liver, where vitamin D microsomal 25-hydroxylase converts it to 25-hydroxy vitamin D (1,25(OH)2D3) [6]. Vitamin D receptors, in particular, are found in various tissues of the human body; indeed, they are almost always expressed in all nucleated elements of our organism [5]. The bioactivity of vitamin D lies not only in the correct functioning of the kidneys, which, as known, convert vitamin D once absorbed by the skin and the gastrointestinal tract into its active form 1,25-dihydroxy vitamin D, due to the enzymatic activity of 1-alpha-hydroxylase, but since this enzyme has been detected in many other tissues, it is reasonable to think that the metabolism of the vitamin or its autocrine and paracrine effects in addition to its endocrine effects are present in various locations in our body [5]. The totality of vitamin D transport depends on the activity of vitamin D binding protein (DBP) and albumin in a measure of 85% and 15%, respectively. Vitamin D is found in the free circulating form in percentages ranging from 0.01% to 3% [5]. Both DBP and albumin are synthesized by the liver, and their production is regulated by estrogens, glucocorticoids, and inflammatory cytokines [5]. It has been observed that the levels of DBP can be lower according to the various breeds, and this is probably linked to the high genetic polymorphism, which determines a common genetic variant and other less common forms.

## 2. Change in Vitamin D Requirements in Women

Observations of an increasing number, mostly coming from animal models, suggest an active role of vitamin D in female reproductive physiology with potential benefits on the correct maturation of women’s sexual functions and with positive repercussions on human reproduction [7,8]. During their life, women undergo various changes in the hormonal and sexual sphere, concerning menarche, possible pregnancies, and breastfeeding, but also the use of contraceptives and, finally, the transition from the period of fertility to menopause [9,10,11,12,13]. Each of these phases presents specific needs and, consequently, sometimes also specific criticalities. Puberty is the transition period from childhood to adulthood and is associated with the onset of menarche and the development of secondary sexual characteristics. In today’s society, during this period, girls tend to have an unbalanced diet, with possible negative repercussions on the regularity of the menstrual cycle and with the appearance of deficiency anemia, sexuality disorders, depression, and reduced growth of bone mass [9,10,11,12,13]. Adolescence is also a very important period for the development of the skeleton, which involves an increase in the need for vitamin D and calcium [9,10,11,12,13]. Conception, implantation, and early pregnancy development, as well as placental development, require energy and micronutrients, including B complex vitamins, vitamins A and D, and folic acid. It should be emphasized that some categories of women are more at risk of nutritional deficiencies [14,15]. Obese women, for example, have an increased risk of vitamin D deficiency; smokers often have lower levels of omega-3 fatty acids in their breast milk, and women who follow a vegetarian/vegan diet instead are exposed to a greater risk of vitamin D deficiency, lack of vitamins B12 and D, and calcium [14,15]. As regards the integrity of the bone system, both in the perimenopausal period and after menopause, vitamin D and calcium supplementation is necessary to counteract the risk of developing osteoporosis and, consequently, the risk of fractures [14,15]. It is, therefore, of fundamental importance to establish the exact role of vitamin D in the function of the various stages of life of the female organism and its role in the development of pathologies related to hormonal status in this category of people [16,17,18,19,20,21,22,23]. Little evidence yet exists on the role of vitamin D in human reproduction, although a role of the latter is easily conceivable above all for the fact that the VDR and 1a-hydroxylase are molecules expressed by both male and female reproductive organs [24]. From an epidemiological point of view, it would seem that the conception rate is greater during the months with prolonged and intense sun exposure and, therefore, with greater production of vitamin D, which also translates into greater ovarian stimulation [25,26].

## 3. The Relationship between Vitamin D and Estrogens

As we have said, vitamin D performs multiple functions in our body and is involved in many diseases. Most of the actions performed by vitamin D are related to its intake and the affinity of its precursors with the vitamin D nuclear receptor (VDR) [27,28]. The VDR is expressed in organs, such as the gut, skeleton, and parathyroid gland in the ovaries and testicles; above all, the function of vitamin D at the level of both sexes is poorly understood [27,28]. It has been observed that vitamin D and estrogen have a double relationship in the sense that low levels of estrogen are associated with falls of osteopenia or rickets, while it has been seen that VDR at the level gonadal plays a role in the production of estrogen [27,28]. It has been showed that vitamin D regulates the biosynthesis of estrogens by means of calcitropic activity and by maintaining the homeostasis of intracellular and extracellular calcium in balance and by acting directly in the regulation of the expression of the aromatase gene [28]. The products of expression of the aromatase gene, such as Cytochrome P450 Family 19 Subfamily A Member 1 (CYP19A1) protein, an aromatase that plays a critical role in estrogen biosynthesis, thus, affects several organ-specific dysfunctions related to body fat distribution and to lipid metabolism regulations. In particular, the CYP19A1 rs10046 polymorphism is considered an important cardiovascular risk factor as it would seem to be associated with the increase in lipoproteins such as circulating apoB with high atherogenic potential, with insulin resistance as a metabolic condition for the onset of type 2 diabetes mellitus and arterial hypertension [29,30,31]. Some studies have observed that estrogens and, in particular, progesterone can reduce the risk of colorectal cancer and the hereditary variation in sex hormone genes may be a mechanism by which sex hormones influence the carcinogenesis of this tumor [32]. This aspect actively affects products that are derived from the aromatase gene. The biological activity of estrogens on the carcinogenic transformation of the colon still remains unclear. Studies on rodents have shown that estrogens, through their binding with the alpha receptor and/or the beta receptor, seem to inhibit the growth of neoplastic cells in the course of colon cancer, as well as the disappearance of both receptors in tissues taken after removal has also been found for colorectal cancer [33,34,35]. It has also been suggested that estrogens and, in particular, estradiol upregulate DNA repair genes in colon cells (mismatch repair) [36,37]. However, other studies have conversely suggested that estrogens stimulate cell proliferation in colon cancer [38,39]. 

The overexpression of aromatase is implicated not only in the pathogenesis of colorectal cancer but also participates in the pathogenesis and growth of those malignant tumors typical of the female gender, such as those of the breast, endometrium, and ovaries [40].

Several cognitive functions are influenced by concentrations of estradiol determined by the activity of the parts of the aromatase. This enzyme is highly expressed in various brain regions and appears to be implicated in potential cognitive deficits, including Alzheimer’s disease, and, in particular, those showing a polymorphism of the CYP19 gene may have prognostically more severe aging from a cognitive–functional point of view [41,42].

Also, in autoimmune pathology with rheumatoid arthritis, genes including IL-1, aromatase, and corticotropin-releasing hormone production are associated with hormonal and reproductive factors, affecting the susceptibility and severity of this disease. In fact, it is possible to observe that this disease is more common in women than men, especially before menopause [43,44].

## 4. Vitamin D, Gender, and Pathological Conditions

Once the link between vitamin D and gender has been established and the importance of estrogen in the functions of this molecule has been observed, we will review any pathological conditions in which there are specific peculiarities for the female sex (Table 1 and Table 2).

## 5. Vitamin D and Reproductive Organ Disorders

There are many pathological conditions of the reproductive organs that could be dependent on the function of vitamin D. 

In the case of endometriosis, the presence of inflammatory and immunological processes has been evoked. It has been seen that in these patients, the 1a-hydroxylase gene is extensively expressed in ectopic endometrial cells [45], as is the case for the receptor VDR with elevated vitamin D values. This would suggest that vitamin D affects the cells and inflammatory cytokines that support inflammation in the course of endometriosis. 

Polycystic ovary syndrome (PCOS) is autoimmune in origin, with a strong genetic component that manifests itself with oligomenorrhea, infertility, and insulin resistance. In these patients, low vitamin D values correlate with a worsening of metabolic manifestations, with a higher incidence of obesity and diabetes [46]. Several studies have shown that polymorphisms involving the VDR receptor are the basis of poor responsiveness to vitamin D [47,48,49], with the possibility that different isoforms of the latter have effects on luteinizing hormone, sex hormone binding globulin levels, and testosterone involved in the pathogenesis of PCOS [47,48,49]. In this regard, the administration of vitamin D in small cases has given an improvement in insulin resistance and a greater regularity of ovulation and lipid profile [50,51].

## 6. Vitamin D and Pregnancy-Related Disorders

The role of vitamin D has also been considered with regard to disorders connected to the pregnancy of gestational diabetes; a relationship between polymorphic forms of vitamin D and impaired glucose tolerance and insulin release has been identified [52]. Furthermore, CYP27B1 polymorphisms that may modulate the action of vitamin D and contribute to insulin resistance have been identified in these patients [53]. These findings are also supported by epidemiological data concerning the concentration of vitamin D at diagnosis, which correlated with the values of the glucose load curve and glycated hemoglobin [53,54]. On the contrary, vitamin D supplementation has brought about an improvement in glycemic values and other metabolic parameters [55].

Also, arterial hypertension in pregnancy and preeclampsia was related to the vitamin D concentration of affected pregnant women. Epidemiologically, it has been seen that the symptoms of preeclampsia show greater incidence and severity during seasons with little solar exposure [56]. From the point of clinical examination, pregnant women affected by preeclampsia show hypovitaminosis D and alterations of calcium metabolism compared to unaffected pregnant women [57]. Furthermore, another study showed that the onset of early preeclampsia was associated with preterm deliveries, with infants small for gestational age and poor growth demonstrating that vitamin D also plays a role in the intrauterine growth of the infant [58]. There are many mechanisms by which hypovitaminosis D contributes to this pathology in pregnancy. Some studies have suggested that vitamin D has a role in the regulation of inflammatory and immunological processes and endothelial dysfunction at the placental level expressing 1a-hydroxylase [59,60] and, in turn, metabolizing its active metabolite 1a,25(OH)2D3 [61] and acting on gene transcription of angiogenic factors such as endothelial growth factor (VEGF) [62,63].

## 7. Metabolic Syndrome, Vitamin D, and Gender

The main metabolic pathologies are linked to mechanisms of either insulin insufficiency or insulin resistance and lead to the various forms of diabetes mellitus and obesity to which cardiovascular diseases are finally correlated. In the case of deficient insulin production, the association with type 1 diabetes mellitus is known, where the destruction of pancreatic beta cells seems to be important above all by means of immune-mediated mechanisms [64]. It has been observed that these cells express the VDR receptor and also the 1-α hydroxylase, which is direct of the genes for aromatase and for which it is conceivable that there is a correlation between vitamin D levels and the insulin response in our body [65]. Furthermore, precisely because of the existing immune reaction underlying the destruction of beta cells, vitamin D would play a role in protecting these cells from the immune reaction against them [66]. A meta-analysis found that patients with type 1 diabetes mellitus show polymorphisms of the VDR gene and that this correlates with a higher metabolic risk [67]. Furthermore, there appears to be an elevated rate of anti-DBP serum antibodies in diabetic patients and, therefore, DBP expressed on pancreatic cells may be a target of regulatory immune T cells [46]. Furthermore, it would seem that the predisposition to type 1 diabetes mellitus derives from low levels of DBP present in the pregnant woman in the third trimester of pregnancy and this correlates to a high metabolic risk for the offspring [68,69]. A beneficial effect of cholecalciferol supplementation has been observed, above all in childhood, on the immune median damage to pancreatic islet cells [6,70,71], while, above all, a prospective study on the environmental determinants of diabetes in young women did not demonstrate advantages in supplementing the diet of pregnant women with vitamin D in preventing immune-mediated damage to the pancreatic islets [72]. In general, many factors influence the prevention or early deficit of insulin production and it is not yet known at what dosage vitamin D can or cannot induce an improvement in the inflammatory damage of the pancreatic islets. What is instead established is that pregnancy or more, generally, the female gender can correlate with a wide variability of circulating vitamin D values, and this represents a risk factor, whatever the pathogenic mechanism, of the onset of type 1 diabetes mellitus in the future unborn child [72,73].

Vitamin D deficiency is considerably associated with obesity at any age and, therefore, there is an increase in the prevalence of type 2 diabetes mellitus, dyslipidemia, and metabolic syndrome [6]. However, this association does not emerge for all populations, as studies involving different populations have not shown any role of vitamin D in improving insulin resistance [74,75]. Although, in all cases examined, both in adults and children, there are no significant differences between vitamin D deficiency and diabetes mellitus, a possible difference in the onset of gestational diabetes and hyperglycemia in pregnancy has recently been hypothesized [76]. Thus, vitamin D concentration may be important in the onset of gestational diabetes through conditioning pancreatic beta-cell function, modulation of insulin sensitivity, and inflammatory status [76,77]. Therefore, it has been shown that the correction of maternal vitamin D deficiency in early pregnancy is associated with lower glycemic values and a lower onset of gestational diabetes mellitus [76,77]. In conclusion, in the current state of knowledge, vitamin D appears to play a role in the onset and management of diabetes mellitus. In gestational diabetes, vitamin D deficiency seems to be related to the typical hormone status of the female gender in the gestational period. Most studies agree that appropriate vitamin D supplementation can prevent and improve all forms of diabetes, but it is difficult to establish adequate times and dosages.

## 8. Autoimmune Diseases, Vitamin D, and Gender

Vitamin D deficiency is also considered a risk factor for the onset and evolution of autoimmune diseases, given its immunoregulatory capacity by mitigating excessive reactivity towards the body’s own cells and promoting immunological tolerance [78]. The role of vitamin D in autoimmune diseases acquires importance from a gender perspective as it is characterized by a propensity towards the female sex [78]. This phenomenon seems to be related to the activity of estrogens, which, as we have seen, influence the metabolism of vitamin D at various levels and act in different ways on the male and female immune systems. In addition to the role of estrogen on the metabolism of vitamin D, it is also useful to underline how the different hormonal structure leads to such differences in the incidence of autoimmune diseases between the two sexes [78,79,80]. In males, there seems to be a greater immunological tolerance, probably linked to the action of androgens; on the contrary, the greater immunoreactivity and the poor tolerance towards the self, typical of the female sex, are probably related to estrogens. This gives women a protective effect against infections but, on the other hand, also a greater tendency to develop autoimmune diseases [78,79,80]. The different nature of the autoimmune response in the two sexes as well as being linked to the different hormonal structure that is able to modulate the Th1/Th2 balance is also related to an interaction between genetic factors and sex hormones [81]. This phenomenon leads us to think that the presentation of the antigen is an important element in defining the immune reactivity of each individual and, therefore, above all, an immunologically very active organ, such as the intestine, based on its microbial composition, can influence the development of these diseases [82,83]. Studies reporting specific gender associations of the composition of the intestinal microbiome show that rather than genetic differences, the different bacterial composition is linked to specific factors such as lifestyle [84]. For the purposes of our discussion, an important question is how sex hormones and specific compositions of the gut microbiome interact in gender autoimmunity and what is the role of vitamin D in this scenario. It has been observed that the different bacterial species of the intestinal microbiome interact with steroids, suggesting an impact of the latter in influencing the intestinal ecosystem and the various bacterial species able to transform intestinal sterols into steroids by promoting their reabsorption through the enterohepatic circulation [85,86]. However, it is not clear whether microbiome-derived steroids have an impact on host immunity. Recent data regarding the influence of gut microbes on gender differences in the immune response suggest that both bacteria and sex hormones directly interact to determine the fate of autoimmune diseases in genetically predisposed individuals [84]. The role of vitamin D in the intestinal microbiota is twofold, i.e., of an immune nature and of a metabolic nature. We have already highlighted the role of vitamin D in modulating the innate immune response and the function of regulatory T cells in antigen recognition [1,2]. It should also be mentioned that the vitamin D system and its receptor VDR regulate the metabolism of numerous molecules present in the intestine, and, in some cases, gender differences have come from animal models in the role of the vitamin D–VDR system. It has been demonstrated in female mice that the female gender is more susceptible to VDR deletion than in male mice, and this would result in an increase in bile acids [87,88], and instead of the male gender type of polyamines [89], multipurpose molecules that regulate gene expression, cell division, differentiation, embryogenesis, senescence, and immune response indicate a specific role of sex-specific hormones in gender differences related to the functioning of the vitamin D/VDR system.

## 9. Cancer, Vitamin D, and Gender

Some cell culture studies have shown that vitamin D prevents uncontrolled cell growth [90,91,92]. Most of the studies on the impact of vitamin D in cancer come from breast studies, the colon, and the prostate. In this regard, epidemiological studies associate vitamin D deficiency with the risk for these three tumors [91,92,93]. In the case of breast cancer, vitamin D would seem to have a role in inhibiting the expression of the tumor progression gene (Id1). It has also been observed that the lack of VDR causes an increase in tumor growth and the development of metastases [91]. One study reported that VDR single-nucleotide polymorphisms, where present, can determine vitamin D deficiency, and following vitamin D supplementation, there is an improvement in prognostic markers of disease [94]. Although the data suggest a protective role of vitamin D, prospective studies seem to disprove this finding since factors, such as obesity, alcohol consumption, and diet, can affect the role of vitamin D [95,96]. However, two other prospective studies [97,98] noted that vitamin D supplementation reduced the risk of postmenopausal breast cancer and improved breast cancer survival in treatment-experienced patients, respectively. The justification for this discrepancy in results could be linked to various factors affecting the action of vitamin D and its receptors, such as lifestyle, diet, and smoking habits, but the role of genetic predisposition seems to preponderate, which means that the action of vitamin D is in the control of tumor progression, as in other activities in our body [97,98].

There are also data that see vitamin D as an agent capable of interfering with colorectal cancer intervening directly on the adenoma–carcinoma progression sequence [99,100]. In contrast, other authors did not find the same results [101,102]. However, despite this evidence, the data do not demonstrate a significant role of vitamin D supplementation in colorectal tumorigenesis and survival after cancer [103,104,105]. These results may be justified by the fact that the real amount of vitamin D necessary to promote the biological prevention of colorectal cancer is not well known. In fact, another study showed how in metastatic cancer, high doses of vitamin D associated with chemotherapy increased disease-free survival by significantly reducing annual mortality rates [106]. Moreover, it also suggests the presence of many confounding factors and genetic roles concerning the VDR polymorphism for this type of tumor, which is still only the subject of study hypotheses. Epidemiological data indicate a better survival of women affected by colorectal cancer, indicating a gender difference, linked to the protective role of the hormone estrogen [107]. We have already stressed that the intestinal microbiota based on its composition can metabolize sterols and, from these, can lead to the production of intestinal estrogens. Some bacteria have the ability to deactivate estrogens by deconjugating them in the presence of β-glucuronidase and transforming them into their active forms [108]. Furthermore, other less protective intestinal bacteria would possess steroid sulfatase and could alter the enterohepatic cycle of estrogenic compounds and contribute to the progression of colorectal carcinogenesis [109]. The immune response is also under the control of the intestinal microbiota, and we have already underlined previously how vitamin D acts in modulating the innate immune response and the function of regulatory T cells in antigen recognition and, at the same time, how the molecule itself is capable of driving intestinal estrogen production through intestinal bacterial species [85,86,110]. In conclusion, gender differences in the incidence and development of colon cancer are mediated by the gene and protein expression of an endocrine nature specific to each sex. The role of estrogens and the dual role played by the vitamin in modulating the metabolic activity of the latter and in regulating the immune response underlie the gender-specific differences.

## 10. Multiple Sclerosis, Vitamin D, and Gender

Among neurodegenerative diseases, multiple sclerosis (MS) has been associated with vitamin D deficiency [111,112,113,114,115]. In general, the cause of this correlation has been attributed by some studies to poor sun exposure, by others from a dysfunction of the melanocortin 1 receptor gene [116], and from others to polymorphism in genes that influence vitamin D levels, especially in children [117]. Especially in women, it was observed that reduced levels of vitamin D were present in those with MS [118]. Other data suggest that good vitamin D levels in mother and child may protect against MS [119]. In the Nurses Health Study, women born to mothers with high vitamin D intake during pregnancy had a reduced risk of MS [120]. These data suggest that in the case of MS, vitamin D plays a greater protective role in the female sex, and in the absence of this or its receptor, there is a high risk of MS, which would already originate from hypovitaminosis D during pregnancy and then be reflected also in the progeny.

## 11. Vitamin D Supplementation in Brain Injury Repair

Vitamin D is a nutrient based on emerging evidence that may also play a role in homeostasis and the reparative recovery of brain injury [120,121]. In addition to the neurodegenerative diseases we have talked about, several studies have investigated the potential benefits of vitamin D in various neurological conditions, including traumatic brain injury and stroke. Vitamin D has been shown to possess anti-inflammatory and neuroprotective properties, which may be helpful in reducing the secondary damage associated with post-traumatic brain injury. Animal studies, however, have shown that vitamin D supplementation has shown promising results in reducing the size of brain lesions by promoting the functional recovery of neurons. In this regard, the combination therapy of vitamin D and progesterone was able to reduce the markers of inflammation and death of neuronal cell trauma brain injury, demonstrating a synergism between the two components and neuroprotective function [121]. Vitamin D in combination with other micronutrients or hormones improved recovery, activity, and protection. Recognized neuroprotective functions of the vitamin include apoptosis of damaged cells and facilitation of myelin repair [120,121]. The presence of vitamin D receptors present in brain tissue leads to the conclusion that vitamin D may have an active role in neuroplasticity and recovery after stroke. In the case of stroke, antioxidant vitamins including vitamin D would seem to play an important role in preventing the production and release of free radicals at the base of endothelial damage, contributing to improvements in motor functions and ambulation in ischemic patients [119,120,121].

## 12. Conclusions

In conclusion, vitamin D can have multiple biological effects on our body, but, in general, studies are slow to unequivocally demonstrate the benefits of any vitamin D supplementation in various pathologies. We have seen how vitamin D plays a mainly protective role in the female gender, especially during fertile life and, in particular, during pregnancy. We have seen how many pathological conditions are reflected in pregnant women to their progeny, and this implies that vitamin D supplementation during pregnancy is an essential element for the health of the mother and the fetus. We have also seen how there is an interchange between vitamin D and estrogen; on the one hand, vitamin D increases the bioavailability of estrogens and, on the other, the latter is able to increase the efficiency of absorption, transport systems, and affinity with its receptor. The important role of the composition of the intestinal microbiota has also been established both in promoting the production of intestinal estrogens with a systemic function and, with them, also a greater absorption of vitamin D. This, especially in women, translates into a greater immune response, which prevents infection and in a greater competence of the regulatory systems of the immune response, which prove to be decisive, above all, in the protection against autoimmune diseases and in the main tumor forms.

## Figures and Tables

**Table 1 biomedicines-11-01762-t001:** Metabolic effects of Vitamin D.

Bone Metabolism	Immune Function	Cancer	Reproductive Disease	Metabolism	Neurological Disease
↑ Bone metabolization	↑ Differentiation of immune cells	↓ Cellular proliferation	PCOS	Diabetes	Multiple sclerosis
↑ Intestinal Ca++ reasorbtion		↓ Angiogenesis	Pre-eclampsia	Obesity	Dementia
↑ Intestinal P+ reasorbtion		↑ Cellular differentiation	Endometriosis	Metabolic syndrome	
↑ Renal Ca++ reasorbtion					
↑ Renal P+ reasorbtion					

**Table 2 biomedicines-11-01762-t002:** Vitamin D deficiency and woman life.

Childhood	Pregnancy	Adulthood	Advanced Aging
Schizopherenia	Pre-eclampsia	Obesity	Dementia
Asthma	Gestational Diabetes	Cancer	Proxymal myopathy
Type 1 diabetes	Bacterial vaginosis	Cardiovascular diseaseas	Osteoporosis
Rickets	Spontaneous Preterms Birth	Multiple sclerosis	Osteomalacia
	Cesarean delivery	Type 2 Diabetes	Frailty

## Data Availability

Not applicable.

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
