# Peer review of "Vitamin D: Can Gender Medicine Have a Role?"

_biomedicines, 2023, doi:10.3390/biomedicines11061762_

Round 1
Reviewer 1 Report
Major comments:
1. The title sounds "catchy" but does not correctly reflect the content of the manuscript. Please revise to a more scientific version.
2. The Abstract is not well written. The authors should explain first the purpose of the review and dissect it than into the different aspects that will be discussed.
3. The manuscript stays in its discussion of what is announced in title and abstract very superficial and is an disappointment for the reader expecting a solid molecular explanation.
4. The selection of references is largely not representative for the field.
5. The manuscript is lacking high quality, illustrative figures of what is discussed in the different sections.
Minor comments:
1. The manuscript should be presented in the journal's template.
2. Please be more precise when you use the term "vitamin D". Is "vitamin D3", "25(OH)D3" or "1,25(OH)2D3" meant. Please do not use different names for the same compound and have a consistent way of writing the names of vitamin D compounds.
3. Please define all abbreviations at first time use and apply them then consistently (e.g. CYP19A1). Please check the whole manuscript.
4. Please be more careful in using terms and abbreviations. For example "VDR receptor" would translate to "Vitamin D receptor receptor".
The manuscript is written very sloppy and needs far more care.
Author Response
Dear Editor in Chief
Biomedicines
Please we send you a revised manuscript entitled VITAMIN D: CAN GENDER MEDICINE HAVE A ROLE? Submitted by Tiziana Ciarambino et al. and we thanks the Reviewers for your crucial comments. We report the manuscript revised with revisison reported in yellow.
REVIEWER 1
Major comments:
- The title sounds "catchy" but does not correctly reflect the content of the manuscript. Please revise to a more scientific version.
R: In accordance with the reviewer's suggestions the title has been changed.
- The Abstract is not well written. The authors should explain first the purpose of the review and dissect it than into the different aspects that will be discussed.
R: In accordance with the reviewer's suggestions, the structure of the abstract has been modified.
- The manuscript stays in its discussion of what is announced in title and abstract very superficial and is a disappointment for the reader expecting a solid molecular explanation.
- The selection of references is largely not representative for the field.
R: In accordance with the reviewer's suggestions, it must be said that the role of gender in the biological functions of vitamin D is a topic that does not yet have solid conceptual foundations. Nonetheless, the choice of bibliographic references was conducted with the criteria set by the publisher and trying to create a rational explanation for the scarce scientific evidence coming from clinical studies.
- The manuscript is lacking high quality, illustrative figures of what is discussed in the different sections.
R: In accordance with the reviewer's suggestions, representative figures of the matter under discussion have been added.
Minor comments:
- The manuscript should be presented in the journal's template.
R: In accordance with the reviewer's suggestions, we had difficulties in paginating the work using the template.
- Please be more precise when you use the term "vitamin D". Is "vitamin D3", "25(OH)D3" or "1,25(OH)2D3" meant. Please do not use different names for the same compound and have a consistent way of writing the names of vitamin D compounds.
R: In accordance with the reviewer's suggestions, the term has been corrected.
- Please define all abbreviations at first time use and apply them then consistently (e.g. CYP19A1). Please check the whole manuscript.
R: In accordance with the reviewer's suggestions, the term has been corrected.
- Please be more careful in using terms and abbreviations. For example "VDR receptor" would translate to "Vitamin D receptor receptor".
R: In accordance with the reviewer's suggestions, the term has been corrected.
Comments on the Quality of English Language
The manuscript is written very sloppy and needs far more care.
R: In accordance with the reviewer's suggestions, the text will subsequently be reviewed in English with the help of the editor.
Best regards
Tiziana Ciarambino
REVIEWER 2
This is an interesting narrative review about the characteristics of vitamin D status and functions within gender and in different pathologies.
I find that the first part (pages 2 to 5) needs reviewing, English is difficult to understand (too long sentences). I have some comments, specially to this first part:
-Abstract, lines 14-16: “The main changes that take place during a woman's life are represented by the rapid development in adolescence with the onset of menarche…”. This sentence is difficult to understand, I suggest rephrasing.
R: In accordance with the reviewer's suggestions, the phrase has been corrected and clarified.
-Page 4, lines 4-7: “Another important factor on which the biological activity of vitamin D depends is the vitamin D binding protein (DBP) plasma protein with a vector function necessary for the transport of vitamin D and its metabolites, as well as an adequate concentration of albumin, is also important.”. This sentence is difficult to understand, I suggest rephrasing.
R: In accordance with the reviewer's suggestions, the phrase has been deleted.
-Page 4, lines 11-13: I suggest adding a reference to this statement.
R: In accordance with the reviewer's suggestions, a reference has been added.
-Page 4, lines 20-23: this sentence is too long and very difficult to read. I suggest rephrasing.
R: In accordance with the reviewer's suggestions, the phrase has been corrected and clarified.
-Page 4, lines 26-30: Sexuality disorders are probably not related to vitamin D deficiency, o this should be correctly supported with references.
R: In accordance with the reviewer's suggestions, the phrase has been corrected and clarified.
-Page 4, lines 26 -27: Girls like men, better women like men.
R: In accordance with the reviewer's suggestions, the phrase has been corrected and clarified.
-Page 5, line 7: Please rephrase, redundant.
R: In accordance with the reviewer's suggestions, the phrase has been corrected and clarified.
Reviewer 2 Report
This is an interesting narrative review about the characteristics of vitamin D status and functions within gender and in different pathologies.
I find that the first part (pages 2 to 5) needs reviewing, English is difficult to understand (too long sentences). I have some comments, specially to this first part:
-Abstract, lines 14-16: “The main changes that take place during a woman's life are represented by the rapid development in adolescence with the onset of menarche…”. This sentence is difficult to understand, I suggest rephrasing.
-Page 4, lines 4-7: “Another important factor on which the biological activity of vitamin D depends is the vitamin D binding protein (DBP) plasma protein with a vector function necessary for the transport of vitamin D and its metabolites, as well as an adequate concentration of albumin is also important.”. This sentence is difficult to understand, I suggest rephrasing.
-Page 4, lines 11-13: I suggest adding a reference to this statement.
-Page4, lines 20-23: this sentence is too long and very difficult to read. I suggest rephrasing.
-Page4, lines 26-30: Sexuality disorders are probably not related to vitamin D deficiency, o this should be correctly supported with references.
-Page 4, lines 26 -27: Girls like men, better women like men.
-Page5, line 7: Please rephrase, redundant.
Author Response
Dear Editor in Chief
Biomedicines
Please we send you a revised manuscript entitled VITAMIN D: CAN GENDER MEDICINE HAVE A ROLE? Submitted by Tiziana Ciarambino et al. and we thanks the Reviewers for your crucial comments. We report the manuscript revised with revisison reported in yellow.
REVIEWER 2
This is an interesting narrative review about the characteristics of vitamin D status and functions within gender and in different pathologies.
I find that the first part (pages 2 to 5) needs reviewing, English is difficult to understand (too long sentences). I have some comments, specially to this first part:
-Abstract, lines 14-16: “The main changes that take place during a woman's life are represented by the rapid development in adolescence with the onset of menarche…”. This sentence is difficult to understand, I suggest rephrasing.
R: In accordance with the reviewer's suggestions, the phrase has been corrected and clarified.
-Page 4, lines 4-7: “Another important factor on which the biological activity of vitamin D depends is the vitamin D binding protein (DBP) plasma protein with a vector function necessary for the transport of vitamin D and its metabolites, as well as an adequate concentration of albumin, is also important.”. This sentence is difficult to understand, I suggest rephrasing.
R: In accordance with the reviewer's suggestions, the phrase has been deleted.
-Page 4, lines 11-13: I suggest adding a reference to this statement.
R: In accordance with the reviewer's suggestions, a reference has been added.
-Page 4, lines 20-23: this sentence is too long and very difficult to read. I suggest rephrasing.
R: In accordance with the reviewer's suggestions, the phrase has been corrected and clarified.
-Page 4, lines 26-30: Sexuality disorders are probably not related to vitamin D deficiency, o this should be correctly supported with references.
R: In accordance with the reviewer's suggestions, the phrase has been corrected and clarified.
-Page 4, lines 26 -27: Girls like men, better women like men.
R: In accordance with the reviewer's suggestions, the phrase has been corrected and clarified.
-Page 5, line 7: Please rephrase, redundant.
R: In accordance with the reviewer's suggestions, the phrase has been corrected and clarified.
Best regards
Tiziana Ciarambino
Dear Editor in Chief
Biomedicines
Please we send you a revised manuscript entitled VITAMIN D: CAN GENDER MEDICINE HAVE A ROLE? Submitted by Tiziana Ciarambino et al. and we thanks the Reviewers for your crucial comments. We report the manuscript revised with revisison reported in yellow.
REVIEWER 2
This is an interesting narrative review about the characteristics of vitamin D status and functions within gender and in different pathologies.
I find that the first part (pages 2 to 5) needs reviewing, English is difficult to understand (too long sentences). I have some comments, specially to this first part:
-Abstract, lines 14-16: “The main changes that take place during a woman's life are represented by the rapid development in adolescence with the onset of menarche…”. This sentence is difficult to understand, I suggest rephrasing.
R: In accordance with the reviewer's suggestions, the phrase has been corrected and clarified.
-Page 4, lines 4-7: “Another important factor on which the biological activity of vitamin D depends is the vitamin D binding protein (DBP) plasma protein with a vector function necessary for the transport of vitamin D and its metabolites, as well as an adequate concentration of albumin, is also important.”. This sentence is difficult to understand, I suggest rephrasing.
R: In accordance with the reviewer's suggestions, the phrase has been deleted.
-Page 4, lines 11-13: I suggest adding a reference to this statement.
R: In accordance with the reviewer's suggestions, a reference has been added.
-Page 4, lines 20-23: this sentence is too long and very difficult to read. I suggest rephrasing.
R: In accordance with the reviewer's suggestions, the phrase has been corrected and clarified.
-Page 4, lines 26-30: Sexuality disorders are probably not related to vitamin D deficiency, o this should be correctly supported with references.
R: In accordance with the reviewer's suggestions, the phrase has been corrected and clarified.
-Page 4, lines 26 -27: Girls like men, better women like men.
R: In accordance with the reviewer's suggestions, the phrase has been corrected and clarified.
-Page 5, line 7: Please rephrase, redundant.
R: In accordance with the reviewer's suggestions, the phrase has been corrected and clarified.
Best regards
Tiziana Ciarambino
Dear Editor in Chief
Biomedicines
Please we send you a revised manuscript entitled VITAMIN D: CAN GENDER MEDICINE HAVE A ROLE? Submitted by Tiziana Ciarambino et al. and we thanks the Reviewers for your crucial comments. We report the manuscript revised with revisison reported in yellow.
REVIEWER 2
This is an interesting narrative review about the characteristics of vitamin D status and functions within gender and in different pathologies.
I find that the first part (pages 2 to 5) needs reviewing, English is difficult to understand (too long sentences). I have some comments, specially to this first part:
-Abstract, lines 14-16: “The main changes that take place during a woman's life are represented by the rapid development in adolescence with the onset of menarche…”. This sentence is difficult to understand, I suggest rephrasing.
R: In accordance with the reviewer's suggestions, the phrase has been corrected and clarified.
-Page 4, lines 4-7: “Another important factor on which the biological activity of vitamin D depends is the vitamin D binding protein (DBP) plasma protein with a vector function necessary for the transport of vitamin D and its metabolites, as well as an adequate concentration of albumin, is also important.”. This sentence is difficult to understand, I suggest rephrasing.
R: In accordance with the reviewer's suggestions, the phrase has been deleted.
-Page 4, lines 11-13: I suggest adding a reference to this statement.
R: In accordance with the reviewer's suggestions, a reference has been added.
-Page 4, lines 20-23: this sentence is too long and very difficult to read. I suggest rephrasing.
R: In accordance with the reviewer's suggestions, the phrase has been corrected and clarified.
-Page 4, lines 26-30: Sexuality disorders are probably not related to vitamin D deficiency, o this should be correctly supported with references.
R: In accordance with the reviewer's suggestions, the phrase has been corrected and clarified.
-Page 4, lines 26 -27: Girls like men, better women like men.
R: In accordance with the reviewer's suggestions, the phrase has been corrected and clarified.
-Page 5, line 7: Please rephrase, redundant.
R: In accordance with the reviewer's suggestions, the phrase has been corrected and clarified.
Best regards
Tiziana Ciarambino
Round 2
Reviewer 1 Report
Despite a revision this manuscript stays superficial and contains a number of factual mistakes
Extensive revision needed
Author Response
Dear Editor in Chief and
Dear Reviewers
Please we send you in attachment a revised manuscript.
REVIEWER 1
- Comments and Suggestions for Authors
Despite a revision, this manuscript stays superficial and contains a number of factual mistakes
R: Taking into account the reviewer's suggestions, and in the spirit of wanting to improve the quality of our work as much as possible, considering the fact that on some points it has only been possible to speculate on the mechanisms underlying gender differences in vitamin D metabolism and function, we would like to better understand from the reviewer what are the gaps in the manuscript. We are convinced that we have thoroughly reviewed all the data in the literature, and if the same reviewer is aware of data that we have not reported due to our negligence, we are pleased to benefit from any suggestion that can improve the work.
- Comments on the Quality of English Language
Extensive revision needed
R: As already reiterated, in accordance with the reviewer's suggestions, the text will subsequently be reviewed in English with the help of the editor.
REVIEWER 2
R: No comments. Thank you for approving our work and welcoming our corrections.
Thank you
Best regards
Tiziana Ciarambino
Reviewer 2 Report
I thank the authors for correnctly adressing my comments.
Author Response
Dear Editor in Chief and
Dear Reviewers
Please we send you in attachment a revised manuscript.
REVIEWER 1
- Comments and Suggestions for Authors
Despite a revision, this manuscript stays superficial and contains a number of factual mistakes
R: Taking into account the reviewer's suggestions, and in the spirit of wanting to improve the quality of our work as much as possible, considering the fact that on some points it has only been possible to speculate on the mechanisms underlying gender differences in vitamin D metabolism and function, we would like to better understand from the reviewer what are the gaps in the manuscript. We are convinced that we have thoroughly reviewed all the data in the literature, and if the same reviewer is aware of data that we have not reported due to our negligence, we are pleased to benefit from any suggestion that can improve the work.
- Comments on the Quality of English Language
Extensive revision needed
R: As already reiterated, in accordance with the reviewer's suggestions, the text will subsequently be reviewed in English with the help of the editor.
REVIEWER 2
R: No comments. Thank you for approving our work and welcoming our corrections.
Thank you
Best regards
Tiziana Ciarambino
Dear Editor in Chief and
Dear Reviewers
Please we send you in attachment a revised manuscript.
REVIEWER 1
- Comments and Suggestions for Authors
Despite a revision, this manuscript stays superficial and contains a number of factual mistakes
R: Taking into account the reviewer's suggestions, and in the spirit of wanting to improve the quality of our work as much as possible, considering the fact that on some points it has only been possible to speculate on the mechanisms underlying gender differences in vitamin D metabolism and function, we would like to better understand from the reviewer what are the gaps in the manuscript. We are convinced that we have thoroughly reviewed all the data in the literature, and if the same reviewer is aware of data that we have not reported due to our negligence, we are pleased to benefit from any suggestion that can improve the work.
- Comments on the Quality of English Language
Extensive revision needed
R: As already reiterated, in accordance with the reviewer's suggestions, the text will subsequently be reviewed in English with the help of the editor.
REVIEWER 2
R: No comments. Thank you for approving our work and welcoming our corrections.
Thank you
Best regards
Tiziana Ciarambino
Dear Editor in Chief and
Dear Reviewers
Please we send you in attachment a revised manuscript.
REVIEWER 1
- Comments and Suggestions for Authors
Despite a revision, this manuscript stays superficial and contains a number of factual mistakes
R: Taking into account the reviewer's suggestions, and in the spirit of wanting to improve the quality of our work as much as possible, considering the fact that on some points it has only been possible to speculate on the mechanisms underlying gender differences in vitamin D metabolism and function, we would like to better understand from the reviewer what are the gaps in the manuscript. We are convinced that we have thoroughly reviewed all the data in the literature, and if the same reviewer is aware of data that we have not reported due to our negligence, we are pleased to benefit from any suggestion that can improve the work.
- Comments on the Quality of English Language
Extensive revision needed
R: As already reiterated, in accordance with the reviewer's suggestions, the text will subsequently be reviewed in English with the help of the editor.
REVIEWER 2
R: No comments. Thank you for approving our work and welcoming our corrections.
Thank you
Best regards
Tiziana Ciarambino
Dear Editor in Chief and
Dear Reviewers
Please we send you in attachment a revised manuscript.
REVIEWER 1
- Comments and Suggestions for Authors
Despite a revision, this manuscript stays superficial and contains a number of factual mistakes
R: Taking into account the reviewer's suggestions, and in the spirit of wanting to improve the quality of our work as much as possible, considering the fact that on some points it has only been possible to speculate on the mechanisms underlying gender differences in vitamin D metabolism and function, we would like to better understand from the reviewer what are the gaps in the manuscript. We are convinced that we have thoroughly reviewed all the data in the literature, and if the same reviewer is aware of data that we have not reported due to our negligence, we are pleased to benefit from any suggestion that can improve the work.
- Comments on the Quality of English Language
Extensive revision needed
R: As already reiterated, in accordance with the reviewer's suggestions, the text will subsequently be reviewed in English with the help of the editor.
REVIEWER 2
R: No comments. Thank you for approving our work and welcoming our corrections.
Thank you
Best regards
Tiziana Ciarambino
Dear Editor in Chief and
Dear Reviewers
Please we send you in attachment a revised manuscript.
REVIEWER 1
- Comments and Suggestions for Authors
Despite a revision, this manuscript stays superficial and contains a number of factual mistakes
R: Taking into account the reviewer's suggestions, and in the spirit of wanting to improve the quality of our work as much as possible, considering the fact that on some points it has only been possible to speculate on the mechanisms underlying gender differences in vitamin D metabolism and function, we would like to better understand from the reviewer what are the gaps in the manuscript. We are convinced that we have thoroughly reviewed all the data in the literature, and if the same reviewer is aware of data that we have not reported due to our negligence, we are pleased to benefit from any suggestion that can improve the work.
- Comments on the Quality of English Language
Extensive revision needed
R: As already reiterated, in accordance with the reviewer's suggestions, the text will subsequently be reviewed in English with the help of the editor.
REVIEWER 2
R: No comments. Thank you for approving our work and welcoming our corrections.
Thank you
Best regards
Tiziana Ciarambino
Dear Editor in Chief and
Dear Reviewers
Please we send you in attachment a revised manuscript.
REVIEWER 2
R: No comments. Thank you for approving our work and welcoming our corrections.
Thank you
Best regards
Tiziana Ciarambino